# Association of macro-level determinants with adolescent overweight and suicidal ideation with planning: A cross-sectional study of 21 Latin American and Caribbean Countries

**Christelle Elia** [1], **Alexis Karamanos**[1], **Alexandru Dregan** [2], **Majella O'Keeffe**[1], **Ingrid Wolfe** [3], **Jane Sandall** [4], **Craig Morgan**[2], **J. Kennedy Cruickshank** [1], **Reeta Gobin** [5], **Rainford Wilks**[6], **Seeromanie Harding** [1,3]*

1 Department of Nutritional Sciences, School of Life Course Sciences, Faculty of Life Sciences & Medicine, King's College London, United Kingdom, 2 Department of Psychological Medicine, Institute of Psychiatry, Psychology and Neuroscience, King's College London, United Kingdom, 3 Department of Population Health Sciences, School of Population Health & Environmental Sciences, Faculty of Life Sciences & Medicine, King's College London, United Kingdom, 4 School of Life Course Sciences, Faculty of Life Sciences & Medicine, King's College London, St Thomas' Hospital, London, United Kingdom, 5 Faculty of Medicine, University of Guyana, Guyana, 6 Faculty of Medicine, University of the West Indies, Jamaica

* seeromanie.harding@kcl.ac.uk

**Data Availability Statement:** The GSHS methods and datasets are publicly available on the WHO website (http://www.who.int/chp/gshs/en/).

## Abstract

### Background

Adolescents and young people (10–24 years old) in the Latin America and the Caribbean (LAC) region represent approximately 25% of the region's population. Since the 2008 global economic crisis, the pace of reduction in poverty and income inequality in the LAC region has stalled. The region is characterised by high levels of inequities and is also vulnerable to many natural disasters. Food systems are changing with increased availability and marketing of packaged and fast foods and sugar-sweetened drinks. Adolescence is a formative phase of the life course with multiple physical, emotional and social changes which can make them vulnerable to health problems. We assess the potential impact of macro-determinants, human and economic development as well as income inequality, on 2 top-ranking regional priorities for adolescent nutrition and mental health, using measures of overweight and suicidal ideation and planning which some have shown to be associated.

### Methods and findings

The Global School-based Health Survey (GSHS) is a nationally representative self-administered, school-based survey. We examined overweight/obesity and suicidal ideation with planning by gross domestic product (GDP) per capita or human development index (HDI) in 10–19-year-old adolescents from 21 LAC countries between 2009 and 2013. Sample sizes varied from 943 in Anguilla to 27,988 in Argentina. A total of 55,295 adolescents had a measure of overweight/obesity status, and 59,061 adolescents reported about suicidal ideation with planning. There was equal representation by sex in the surveys (52% girls and 48% boys). A total of 28.8% of boys and 28.1% of girls had overweight/obesity, and 7.5% of boys

**Funding:** CE, MOK received funding from the Medical Research Council (MR/R022739/1). SH, JKC, RG, RW, AK received funding from the Medical Research Council (MR/N015959/1). The funders had no role in study design, data collection and analysis, decision to publish, or preparation of the manuscript.

**Competing interests:** The authors have declared that no competing interests exist.

**Abbreviations:** AA-HA!, Accelerated Action for the Health of Adolescents; BMI, body mass index; CI, confidence interval; GDP, gross domestic product; GNI, gross national income; GSHS, Global School-based Health Survey; HDI, human development index; LAC, Latin America and the Caribbean; LMIC, low- and middle-income country; MICE, multiple imputation chained equations; NCD, noncommunicable disease; OR, odds ratio; SD, standard deviation; SDG, sustainable development goal; SEDLAC, Socio-Economic Database for Latin America and the Caribbean; SSB, sugar-sweetened beverages; STROBE, Strengthening the Reporting of Observational Studies in Epidemiology; WIID, World Inequality Income Database.

and 17.5% of girls reported suicidal ideation with planning over the last 12 months. Adjusted for individual socioeconomic and risk behaviours, and relative to the highest GDP per capita tertile, the middle tertile was associated with 42% (95% confidence interval (CI) 59% to 17%, $p = 0.003$) and 32% (95% CI 60% to 5%, $p = 0.023$), and the lowest tertile with 40% (95% CI 55% to 19%, $p = 0.001$) and 46% (95% CI 59% to 29%, $p < 0.001$) lower chances of overweight/obesity for girls and boys, respectively. A similar positive effect was seen with HDI, with lowest chances of overweight in the lowest tertile compared with the highest tertile for both sexes. Overweight/obesity was positively related with suicidal ideation with planning for girls (odds ratio (OR) 1.12, 95% CI 1.02 to 1.22, $p = 0.009$) and weakly related for boys (OR 1.09, 95% CI 0.96 to 1.24, $p = 0.182$). In contrast to overweight/obesity status, suicidal ideation with planning was not related to macro-level indices despite both outcomes sharing common individual socioeconomic and risk behaviour correlates. Limitations include the dominance of Argentinians in the sample (40%), the exclusion of vulnerable adolescents who dropped out of school, and reporting bias due to stigma of mental health–related issues.

## Conclusions

This study shows that economic and human development were positively associated with adolescent overweight/obesity but not with suicidal ideation with planning. We also observed an interconnectedness between overweight/obesity and suicide ideation with planning among girls. These findings highlight the importance of strategies that engage with both upstream and downstream determinants to improve adolescent nutrition and mental health.

---

## Author summary

### Why was this study done?

- Adolescents and young people (10–24 years old) in the Latin America and the Caribbean (LAC) region represent over 1 quarter of the region's population.

- Adolescent nutrition and mental health are key policy priorities in the region.

- Despite the considerable diversity across the LAC region in economic development, welfare and health systems, little is known about the influence of level of national development or country-level income inequality on adolescent with overweight and obesity or suicidal ideation and planning.

### What did the researchers do and find?

- Using data from 21 LAC countries between 2009 and 2013, we found that adolescents living in more economically developed LAC countries were at higher risk of overweight and obesity.

- Suicidal ideation and planning was not associated with the level of development, but it was associated with overweight and obesity.

## What do these findings mean?

- This study shows that the risk of overweight and obesity is not only limited to individual or household level influences, but also to wider societal influences.

- Countries in the LAC region need to monitor and prevent adolescent overweight/obesity as they transition from lower to middle- and higher-income status.

## Introduction

In 2015, the population of adolescents and young people (10–24 years old) in the Latin America and the Caribbean (LAC) region was estimated to be 26% of the total population in the region [1]. Although income inequality has fallen in recent years, Latin America remains the most unequal region in the world [2]. Since the 2008 global economic crisis, the pace of reduction in poverty and income inequality in the LAC region has stalled [3]. Despite some progress in the country-specific responses to adolescent and youth health, adolescent mortality rates remained constant or worsened, and their risk factors are highly prevalent and social their determinants influence which groups are affected the most [1]. The region is also vulnerable to many natural disasters, and the slow recovery of infrastructure and services disproportionately affects the poorest in rural areas. We assess the potential impact of macro-determinants, human and economic development, on 2 top-ranking regional priorities for adolescent nutrition and mental health, overweight, and suicidal behaviour, which some studies have shown to be related [1,4,5].

Several global and LAC initiatives foster the importance of adolescent health, including The United Nations' Global Strategy for Women's, Children's and Adolescents' Health 2016–2030, WHO's Global Accelerated Action for the Health of Adolescents (AA-HA!), inclusion of adolescents in 12 of 232 sustainable development goals (SDGs) indicators and in the LAC "Every Woman, Every Child" movement [6,7]. The youth health research agenda in the region includes a focus on both mental health and nutrition [1], the notion that the reduction of health inequalities is a responsibility of society at large and that a wide set of actors and institutions are necessary to achieve a more equal and healthy society. However, regional- and country-level initiatives have not yet translated into major health gains for young people. There are some policies addressing undernutrition and micronutrient deficiencies [8], but there is a general lack of comprehensive and intersectoral policies and good quality and timely data to track progress in adolescent health [9].

The prevalence of malnutrition, undernutrition, and excess weight varies across the region. The prevalence of overweight and obesity in LAC is reaching the levels of high-income countries [10,11] and projected to increase by 5% among girls over the next decade [11]. Among 10- to 19-year olds, the prevalence of undernutrition is highest in Guyana and Trinidad with levels as high 19% to 20%, and the prevalence of overweight and obesity is highest in Argentina, Bahamas, Chile, and Mexico (34%). Excess weight, a risk factor for hypertension, diabetes, and a range of other noncommunicable diseases (NCDs), has serious implications for

economic growth in transitioning countries [12]. Trade liberalisation since the 1990s is thought to have had a major impact on food systems with increased availability, advertising and marketing of packaged and fast foods and sugar-sweetened drinks, and similarly advertising and marketing on food choices [13].

Worldwide, suicide is 1 the leading causes of death in youth which accounts for an estimated 6% of all deaths among young people [14,15]. A recent review of adolescent suicidal behaviours in low- and middle-income countries (LMICs) showed that across the LAC region, physical attacks, bullying, loneliness, lack of parental support, and alcohol and tobacco use were strongly related to suicidal behaviours. Adolescent suicide rates are highest in Guyana (18.3 per 100,000) and lowest in Jamaica (1.2/100,000) [15]. Trends in deaths from self-harm and inter-personal violence have been increasing at different rates across the region, between 1990 and 2017, with the highest increase among 15- to 19-year olds in Latin America. As in high-income countries, structural determinants such as economic poverty are associated with suicide-related behaviours. Current alcohol use and lack of peer support are risk factors, while supportive parenting is a key protective factor—reducing the prevalence of suicidal ideation with planning by up to a third [2,15,16].

Overweight, obesity, and suicide-related behaviours are among the most prevalent and debilitating physical and mental conditions affecting young people [17,18], with the 2 conditions sharing common risk factors (poverty, mental illness, and early life adverse events) which point towards similar macroeconomic determinants [14,19,20]. Systematic reviews show conflicting evidence of the association between overweight/obesity and suicide. In their review of 15 studies, Amiri and Behnezhad reported an inverse association between overweight/obesity with suicide-related mortality and suicidal ideation with planning, but a positive association between overweight/obesity and with suicidal ideation with planning [4]. A possible bidirectional association was noted with overweight/obesity being associated with an increased risk of suicidal ideation with planning, while drugs used to treat severe mental illness were strongly associated with weight gain. Another systematic review by Perera and colleagues showed that underweight was significantly associated with an increased risk of completed suicide, and obesity and overweight were significantly associated with a decreased risk of completed suicide relative to normal weight [21]. There were only 2 adolescent studies in these reviews, both of which showed a positive association with suicidal ideation with planning, 1 with overweight [22,23].

There is some evidence of a relationship between increased country-level income inequality and diminished child well-being among affluent countries in Europe, the United States of America, and Australia [24]. However, little is known about the association between the level of national development, country-level income inequality, overweight, and suicidal ideation with planning in adolescence in the LAC, despite the considerable diversity across the region in economic development, welfare, and health systems. This is an important question for developing effective policies, not least for economic and social support. We used the Global School-based student Health Surveys (GSHS) for 21 LAC countries to examine (i) the association between national indices (economic growth, human development, and inequality) and overweight/obesity and suicidal ideation and planning in boys and girls; and (ii) the association between overweight/obesity and suicidal ideation with planning in boys and girls, after adjustment for individual level risk factors.

## Methods

The GSHS is a nationally representative self-administered, school-based survey conducted by LMICs, jointly developed by WHO and the US Centre of Disease and Control and Prevention,

**Table 1. Survey year, sample size, GDP (per capita, US$), HDI, and Gini index for countries in LAC that participated in the GSHS[1], 2009–2013.**

| Country | Income classification[a] | Year of survey | Overall response rate | Sample size | GDP/capita[b] | HDI[c] | Gini[d] |
|---|---|---|---|---|---|---|---|
| Argentina | UMIC | 2012 | 71% | 27,988 | 12,970 | 0.822 | 40.0 |
| Anguilla | UMIC | 2009 | 84% | 943 | 19,469 | N/A | N/A |
| Antigua and Barbuda | UMIC | 2009 | 67% | 1,244 | 13,012 | 0.782 | N/A |
| Chile | UMIC | 2013 | 60% | 2,033 | 15,941 | 0.841 | 47.2 |
| Costa Rica | UMIC | 2009 | 72% | 2,669 | 6,809 | 0.749 | 50.2 |
| Dominica | UMIC | 2009 | 84% | 1,638 | 6,866 | 0.721 | 44.4 |
| Jamaica | UMIC | 2012 | 72% | 1,748 | 5,210 | 0.727 | 55.4 |
| St Kitts Nevis | UMIC | 2011 | 70% | 1,722 | 14,483 | 0.746 | 40.0 |
| Suriname | UMIC | 2009 | 89% | 1,692 | 7,444 | 0.704 | 45.0 |
| Uruguay | UMIC | 2012 | 77% | 3,488 | 15,092 | 0.788 | N/A |
| Belize | LMIC | 2011 | 88% | 2,102 | 4,516 | 0.702 | N/A |
| Bolivia | LMIC | 2012 | 88% | 3,535 | 2,645 | 0.661 | 46.25 |
| El Salvador | LMIC | 2013 | 84% | 1,883 | 3,896 | 0.676 | 43.4 |
| Guatemala | LMIC | 2009 | 81% | 5,511 | 2,636 | 0.609 | 54.9 |
| Guyana | LMIC | 2010 | 76% | 2,362 | 3,026 | 0.624 | N/A |
| Honduras | LMIC | 2010 | 79% | 1,595 | 1,933 | 0.611 | 52.3 |
| Peru | LMIC | 2010 | 84% | 2,864 | 5,022 | 0.721 | 45.3 |
| Bahamas | HIC | 2013 | 78% | 1,350 | 22,590 | 0.789 | 41.75 |
| Barbados | HIC | 2011 | 73% | 1,625 | 15,534 | 0.785 | 45.0 |
| British Virgin Islands | HIC | 2009 | 90% | 1,655 | 33,119 | N/A | N/A |
| Trinidad and Tobago | HIC | 2011 | 90% | 2,782 | 19,054 | 0.772 | 39,1 |

GDP, gross domestic product; GSHS, Global School-based Health Survey; HDI, human development index; HIC, high-income countries; LAC, Latin America and the Caribbean; LIC, low-income country; LMC, lower-middle income country; UMC, upper-middle income country; corresponding to the year of survey; UNDP, United Nations Development Programme.

a Classification according to the World Bank.

b GDP per capita based on the World Bank Group data (https://data.worldbank.org).

c HDI based on UNDP data (http://hdr.undp.org/en/data), composite index of life expectancy, education and per capita income indicator, which is used to rank countries from 0 to 1 (the higher the better).

d Gini based on WIID (https://www4.wider.unu.edu/). A higher Gini index indicates higher economic inequality. Values on the Gini index represent the following years for participating countries: Argentina, 2012; Chile, 2013; Costa Rica, 2009; Dominica, 2008; Jamaica, 2012; St Kitts and Nevis, 2009; Suriname, 2008; Uruguay, 2012; Bolivia, 2012; El Salvador, 2013; Guatemala, 2013; Honduras, 2010; Peru, 2010; Bahamas, 2013; Barbados, 2010.

in collaboration with UNICEF, UNESCO, and UNAIDS. The GSHS datasets are publicly available; details of the study and questionnaires can be found on WHO website (http://www.who.int/chp/gshs/en). An analysis plan (see S1 Analysis Plan) was prepared in January 2019, whereby the authors decided not to include Argentina in the analyses because a large sample size would skew the results. However, in agreement to reviewers' comments that meta-analytic type methods are designed to deal with different samples sizes, we subsequently included Argentina in our revised manuscript. Our cross-sectional analyses used the most recent GSHS data from nationally representative samples in 21 LAC countries (N = 72,409) between 2009 and 2013 (Table 1). Response rates varied from 60% in Chile to 90% in British Virgin Islands and Trinidad and Tobago. Four countries were classified as high-income countries but were included given the scant information on adolescent health behaviours in high-income countries that are not Organisation for Economic Co-operation and Development countries. This study is reported as per the Strengthening the Reporting of Observational Studies in Epidemiology (STROBE) guideline (S1 STROBE Checklist).

## Outcomes

**Overweight/Obesity and suicide ideation with planning.** Table 2 shows the relevant questions from the GSHS. Outcomes refer to overweight and suicidal ideation with planning. The GSHS contained variables for underweight, overweight, and obesity. Each student's body mass index (BMI) was assessed paper by the surveys' researchers and recorded on slips of paper that were given to each student. Students then entered height and weight on the GSHS answer sheet. The underweight category of BMI was derived using international age- and sex-specific child BMI cut points ($> -2$ standard deviations from the median for age and sex based on WHO growth reference curves). The overweight/obesity category of BMI reflects a BMI cut point of $> +1$ standard deviation from the median for age and sex based on WHO growth reference curves. The main statistical analyses focused on the association between macroeconomic indicators and overweight/obesity as the there was a small number of underweight participants. As underreporting of suicidal attempts is plausible and suicide planning without suicide ideation could be affected by reporting bias [25], we derived a variable that measured suicide ideation with planning to give a binary variable ("yes" if an affirmative response to both ideation and planning; otherwise classified as "no").

## Exposures

**National indices of development and income inequality.** Country- and year-specific macro-indices of gross domestic product (GDP/capita) and the human development index (HDI) at survey time were merged with GSHS data at country level. GDP/capita, a measure of a country's economic output that accounts for its number of people, divides the country's GDP by its total population. It is a measure of a country's standard of living. GDP/capita was extracted from the World Bank Group data (https://data.worldbank.org). The HDI is a composite index of life expectancy, education, and per capita income. It is used to rank countries from 0 to 1 (the higher, the better) for human development and was extracted from http://hdr.undp.org/en/data. The 2 indices were then recoded into tertiles for ease of interpretation, with the high GDP/HDI tertile being the reference category. Information on income inequality was extracted from the World Inequality Income Database (WIID; https://www4.wider.unu.edu/) which includes data from the Socio-Economic Database for Latin America and the Caribbean (SEDLAC), a harmonised set of 6 indicators based on a collection of surveys. The SEDLAC is considered as 1 of the 2 main references for cross-country inequality comparisons in the region. The Gini coefficient was extracted and merged to the GSHS at survey time or closer to survey time at country level. The Gini coefficient measures the inequality among values of a frequency distribution (levels of income). When multiple observations for Gini were available for a country at survey year or a year close to survey time, an average of high-quality observations as judged by WIID was used [26]. A Gini coefficient of 0 expresses perfect equality, where all values are the same, while a Gini coefficient of 1 expresses maximal inequality among values (e.g., for a large number of people where only 1 person has all the income or consumption and all others have none, the Gini coefficient will be nearly 1). For the main analysis, the Gini index was recorded into tertiles, with the high Gini tertile being the reference category.

## Covariates

Covariates included age and psychosocial factors such as bullying, having close friends, feeling lonely, and parental support. Lifestyle behaviours such as cigarette smoking and alcohol use 30 days before the time of the survey were considered as covariates. Food insecurity was used as a proxy measure of socioeconomic circumstances. A detailed description of the covariates used can be found in Table 2.

**Table 2. GSHS[1] questions used in the analysis of adolescent with overweight/obesity and suicidal behaviour in LAC countries, 2009–2013.**

| Variable | Question | Values |
|---|---|---|
| BMI[d] | Percentage of students who measure > +1 SD from the median for age and sex based on WHO growth reference curves<br>Percentage of students who measure > −2 SD from the median for age and sex based on WHO growth reference curves | 0 = normal weight<br>1 = overweight/obesity<br>2 = underweight |
| Suicidal ideation | During the past 12 months, did you ever seriously consider attempting suicide? | 0 = no<br>1 = yes |
| Suicidal planning | During the past 12 months, did you ever make a plan about how you would attempt suicide? | 0 = no<br>1 = yes |
| Age | How old are you? | 0 = ≤ 12 years<br>1 = 13 years<br>2 = 14 years<br>3 = 15 years<br>4 ≥ 16 years |
| Sex | What is your sex? | 0 = boy<br>1 = girl |
| Physical attacks[a] | During the past 12 months, how many times were you physically attacked? | 0 = 0 times<br>1 = 1 time<br>2 = 2 or more times |
| Bullying[b] | During the past 30 days, on how many days were you bullied? | 0 = 0 days<br>1 = 1 or 2 days<br>2 = 3 or more days |
| Food insecurity | During the past 30 days, how often did you go hungry because there was not enough food in your home? | 0 = never or rarely/ sometimes<br>1 = most of the time/always |
| Loneliness | During the past 12 months, how often have you felt lonely? | 0 = never<br>1 = rarely/sometimes<br>2 = most of the time/always |
| Parental support | During the past 30 days, how often did your parents or guardians understand your problems and worries? | 0 = most of the time/always<br>1 = sometimes<br>2 = never/ rarely |
| Few close friends | How many close friends do you have? | 0 = 3 or more<br>1 = 1 or 2<br>2 = none |
| Alcohol use[c] | During the past 30 days, on how many days did you have at least 1 drink containing alcohol? | 0 = 0 days<br>1 = 1 or 2 days<br>2 = 3 or more days |
| Cigarette smoking | During the past 30 days, on how many days did you smoke cigarettes? | 0 = 0 days<br>1 = 1–5 days<br>2 = 6 or more days |

a In the survey questionnaire, physical attack is defined as "when one or more people hit or strike someone, or when one or more people hurt another person with a weapon (such as a stick, knife, or gun)." The survey questionnaire specifies that "it is not a physical attack when two students of about the same strength or power choose to fight each other."

b Bullying is defined as "when a student or group of students say or do bad and unpleasant things to another student," "when a student is teased a lot in an unpleasant way," or "when a student is left out of things on purpose." The survey questionnaire specifies that "it is not bullying when two students of about the same strength or power argue or fight or when teasing is done in a friendly and fun way."

c Drinking alcohol also includes consuming locally produced alcoholic drinks. The survey questionnaire specifies that "drinking alcohol does not include drinking a few sips of wine for religious purposes. A drink is defined as a glass of wine, a bottle of beer, a small glass of liquor, or a mixed drink."

d Based on WHO growth reference curves for 5–19 years old (https://www.who.int/growthref/en/).

BMI, body mass index; GSHS, Global School-based Health Survey; LAC, Latin America and the Caribbean; SD, standard deviation; WHO, World Health Organization.

## Moderators

All analyses were stratified by sex, a key determinant of overweight/obesity and suicidal ideation with planning [27,28].

## Statistical analysis

The dataset for each country included a weighting variable that allows the results to be generalised to the entire population of students. The weighting formula used is $W = W_1 \times W_2 \times f_1 \times f_2 \times f_3$, where $W_1$ corresponds to the inverse probability of selecting each school, and $W_2$ refers to the inverse probability of selecting each classroom, while $f_1$ refers to a school-level nonresponse adjustment factor, $f_2$ to a student-level nonresponse adjustment factor calculated by classroom, and $f_3$ to a post-stratification factor calculated by sex within grade. To account for missing data, we performed multiple imputations using multiple imputation chained equations (MICE) procedure in STATA 15 (StataCorp, College Station, TX, USA), under the missing at random assumption [21]. Item nonresponse ranged from 0.6% for age to 31.4% for cigarette smoking. Thirty multiple imputed datasets were then generated. Due to absent data, Anguilla and Antigua and Barbuda were not included in the analysis of overweight/obesity, and Chile and Barbados in the analysis on suicidal ideation with planning. At the analysis stage, imputed values on the outcomes of interest were dropped, and all estimates were combined using Rubin's rules [29]. Adjusted for clustering within countries, we estimated sex-specific predictive probabilities of overweight/obesity and suicidal ideation with planning in each country using age-adjusted logistic regressions. Sex-specific random-effects logistic regression models were used to estimate odds ratio (OR) and 95% confidence intervals (CIs) for the associations between national indices (HDI, GDP/capita, and Gini index) and overweight/obesity and suicidal ideation with planning unadjusted (Model 1) and adjusted for individual risk factors (Model 2). Random-effects modelling considers the hierarchical structure of the data by clustering students' responses (Level 1) within participating countries (Level 2). This was done in STATA by declaring the hierarchical nature of the GSHS data with the xtset command with student responses in the survey nested within countries. The percentage of between-country differences in overweight/obesity and suicidal ideation with planning after adjustment for GDP/HDI/Gini index (Model 1) and individual risk factors (Model 2) is represented by intraclass correlation coefficients. Adjusted random-effects logistic regression models were also applied to estimate ORs and for the association between overweight/obesity and suicidal ideation with planning. A $p$-value of less than 0.05 was statistically significant in random-effects models. In sensitivity analyses, we applied random-effects linear regression by using a standardised continuous BMI score to correct for positive skewness of BMI as well as continuous macroeconomic indicators of development and economic inequality to minimise both type I and type II errors and increase the precision of estimates. Only linear effects were modelled, as quadratic effects of macroeconomic indicators were not statistically significant. To explore whether macroeconomic indicators were associated with a higher or lower risk of underweight and overweight/obesity compared with normal category of BMI, we applied multinomial logistic regression, adjusted for the nonindependence of observations within countries using the Huber–White variance estimator.

## Ethics statement

Consistent with the GSHS study protocol, questionnaires were administered to all eligible participants in an anonymous, voluntary manner. Parental consent was obtained. Written approval was also obtained from each participating school and from all classroom teachers.

Participant countries in the GSHS follow a standard protocol for ethics approval from the country-specific Ministry of Education, sampling, surveying, and data management.

## Results

Table 3 shows the key characteristics of the total sample. The prevalence of overweight/obesity was similar among boys and girls. Compared with boys, girls reported higher levels of suicidal ideation with planning. They also reported more loneliness, fewer physical attacks, less parental support, smoking, alcohol consumption, and food insecurity. S1 Table shows the distribution of all variables by country. The highest levels of overweight/obesity were observed in the Bahamas and Chile, and suicide ideation with planning in Guyana, Dominica, Peru, and Honduras.

Fig 1 shows the country- and sex-specific predictive probabilities for overweight/obesity by GDP/capita, adjusted for age. A positive relationship between overweight/obesity and GDP/capita was evident for both boys and girls. Similar positive associations were observed for HDI (S1 Fig). However, the associations with the Gini index varied by sex, with a positive association for boys and a negative association for girls (S2 Fig). The corresponding data for suicidal ideation with planning are shown in S3–S5 Figs. Suicidal ideation with planning did not vary markedly by GDP/capita, HDI, or Gini index.

Table 4 shows the association between national indices of development, economic inequality, and overweight/obesity, with additional adjustments for covariates. The intraclass correlation coefficient in the last row of Table 4 shows that adjustment for individual risk factors explained only a small proportion of between country variance. The main effects for national indices show that compared with high GDP per capita tertile, the middle tertile was associated with 42% and 32% and the low tertile with 40% and 46% lower chances of overweight/obesity for girls and boys, respectively. Compared with high HDI tertile, the low tertile was associated with 49% (girls) and 53% (boys) lower chances of overweight/obesity.

S2 and S3 Tables show the results for all covariates in the models. Analyses with GDP per capita showed that bullying was associated with a lower likelihood of overweight/obesity for boys, while frequent and cigarette smoking was associated with a higher likelihood of overweight/obesity among girls.

Table 5 shows no association between levels of national development and economic inequality and suicidal ideation with planning in both boys and girls. Being lonely most of the time for both boys and girls and low food insecurity among boys were independent correlates of suicidal ideation with planning. All psychosocial factors were independent correlates of suicidal ideation with planning for both boys and girls (see S4 and S5 Tables). Increasing age was associated with a higher likelihood of suicidal ideation with planning only among girls, while food insecurity was associated with a higher likelihood of suicidal ideation with planning only among boys.

### Overweight/obesity and suicidal ideation

There was a positive association between overweight/obesity and suicidal ideation with planning for girls (OR 1.12, 95% CI 1.02 to 1.24, $p = 0.009$) after adjusting for age, psychosocial factors, health-related behaviours, and food insecurity. A weak positive association was found for boys (1.09, 95% CI 0.96 to 1.24, $p = 0.182$) after statistical adjustment for covariates.

### Sensitivity analyses

S6 Table shows that among boys, a US$1,000 per capita increase was associated with a 0.02 (95% CI 0.02 to 0.03, $p < 0.001$) standard deviation (SD) in BMI, a unit increase in HDI with a

**Table 3. Sample characteristic % (95% CIs)[†].**

| | Girls | Boys |
|---|---|---|
| *Outcomes* | | |
| Underweight | 1.2 (1.0, 1.5) | 1.7 (1.4, 2.0) |
| Normal weight | 70.7 (68.2, 73.1) | 69.5 (67.3, 71.6) |
| Overweight/obesity[+] | 28.1(25.6, 30.1) | 28.8 (26.8, 31.0) |
| Suicide ideation with planning[++] | 17.5 (16.3, 18.9) | 7.5 (6.9, 8.1)* |
| *Sociodemographic factors* | | |
| *Age* | | |
| <12 years old | 5.6 (4.5, 7.0) | 6.1 (4.8, 7.8) |
| 13 years old | 20.9 (19.1, 22.7) | 20.9 (19.3, 22.7) |
| 14 years old | 27.4 (25.8, 29.0) | 29.0 (27.3, 30.8) |
| 15 years old | 26.9 (25.2, 28.8) | 26.5 (24.6, 28.4) |
| >16 years old | 19.2 (17.7, 20.9) | 17.5 (15.9, 19.2) |
| *Psychosocial factors* | | |
| *Loneliness* | | |
| Never | 56.7 (54.9, 58.6) | 72.4 (70.7, 74.0)* |
| Sometimes | 29.4 (28.0, 30.8) | 21.0 (19.4, 22.6)* |
| Always | 13.8 (13.0, 14.8) | 6.6 (6.1, 7.3)* |
| *Close friends* | | |
| 3 or more | 66.8 (65.0, 68.3) | 70.9 (69.3, 72.4) |
| 1 or 2 | 27.5 (26.0, 29.0) | 22.2 (20.9, 23.6)* |
| None | 5.9 (5.4, 6.4) | 6.8 (6.3, 7.4) |
| *Parental understanding* | | |
| Always | 43.0 (41.4, 44.5) | 41.5 (39.7, 43.3) |
| Sometimes | 18.7 (17.5, 20.1) | 20.9 (19.8, 22.0) |
| Rarely | 38.3 (37.1, 39.5) | 37.6 (36.2, 39.1) |
| *Times bullied (past 30 day)* | | |
| 0 days | 73.0 (69.9, 76.0) | 72.3 (0.70, 0.75) |
| 1 or 2 days | 18.6 (16.4, 21.0) | 18.3 (16.5, 20.3) |
| 3 or more | 8.4 (7.4, 9.5) | 9.4 (8.3, 10.5) |
| *Times attacked (past 12 months)* | | |
| None | 73.0 (69.9, 76.0) | 72.3 (0.70, 0.75) |
| 1 time | 18.6 (16.4, 21.0) | 18.3 (16.5, 20.3) |
| 2 or more times | 8.4 (7.4, 9.5) | 9.4 (8.3, 10.5) |
| *Lifestyle factors* | | |
| *Smoking days (past 30 days)* | | |
| 0 days | 81.9 (80.0, 83.6) | 77.9 (76.0,79.7)* |
| 1 or 2 days | 11.3 (10.3, 12.5) | 14.1 (12.9, 15.5)* |
| 3 or more | 6.8 (5.7, 8.1) | 8.0 (6.8, 9.3) |
| *Alcohol consumption (past 30 days)* | | |
| 0 days | 67.1 (65.3, 68.7) | 64.7 (62.5, 66.8) |
| 1 or 2 days | 19.5 (18.4, 20.7) | 18.3 (17.0, 19.6) |
| 3 or more | 13.4 (12.3, 14.6) | 17.1 (15.5, 18.7)* |
| *Food security* | | |
| *Times gone to bed hungry* | | |
| Never | 84.3 (82.6, 85.8) | 83.0 (81.2, 84.7) |
| Sometimes | 12.5 (11.2, 13.9) | 12.6 (11.3, 14.0) |

(*Continued*)

**Table 3.** (Continued)

|  | Girls | Boys |
|---|---|---|
| Always | 3.3 (2.8, 3.7) | 4.4 (3.8, 5.1) |

† Sample characteristics are based on complete cases.

+ Proportions for weight are based on 26,564 boys and 28,731 girls.

++ Proportions for suicidal behaviour are based on 28,484 boys and 30,577 girls.

CI, confidence interval.

0.02 (95% CI 0.01 to 0.02, $p < 0.001$) SD increase, while a unit increase in the Gini index was associated with a 0.04 (95% CI 0.03 to 0.05, $p < 0.001$) SD decrease in BMI, after adjustment for covariates. For girls, US$1,000 per capita increase was associated with 0.02 (95% CI 0.01 to 0.03, $p < 0.001$) SD increase in BMI, a unit increase in HDI with a 0.01 (95% CI 0.01 to 0.02, $p = 0.024$) SD increase, while a unit increase in Gini was associated with a 0.03 (95% CI 0.02 to 0.04, $p < 0.001$) SD increase in BMI, after adjustment for covariates. Associations between macroeconomic indicators and underweight were not evident (see S7 Table).

## Discussion

To our knowledge, this is the first known attempt to examine the association between indices of national development and adolescent overweight/obesity and suicidal ideation with planning in LAC countries. Lower levels of national development were consistently associated with a lower likelihood of overweight/obesity for both boys and girls. Overweight/obesity was positively associated with suicidal ideation with planning, but the latter was not associated with national indices. Individual level factors such as age, feelings of loneliness, bullying (girls), and food insecurity (boys) were common correlates of both overweight/obesity and suicidal ideation with planning. The prevalence of overweight/obesity was generally similar across sexes, but suicidal ideation with planning was consistently higher in girls than in boys. Among boys, food insecurity in less developed countries was associated with a decreased likelihood of overweight/obesity compared with food insecurity in more developed countries. We highlight both the important upstream and downstream determinants that influence adolescent overweight/

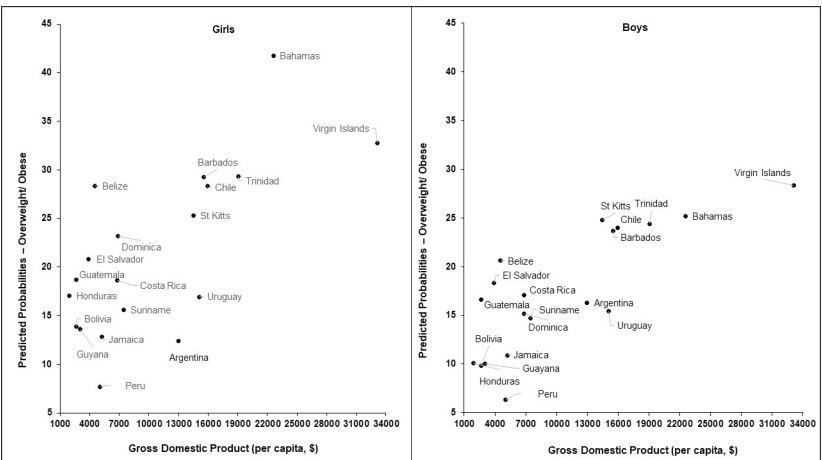

**Fig 1. Age-adjusted predicted probabilities of overweight/obesity by GDP/capita for boys and girls.** GDP, gross domestic product.

**Table 4. Association between national indices of development and with overweight/obesity, adjusted for individual risk behaviours, by sex (N = 55,295).**

| | Girls | Boys | Girls | Boys | Girls | Boys |
|---|---|---|---|---|---|---|
| | OR (95% CI) (p-value) | OR (95% CI) (p-value) | OR (95% CI) (p-value) | OR (95% CI) (p-value) | OR (95% CI) (p-value) | OR (95%CI) (p-value) |
| Higher HDI tertile (ref.) | | | | | | |
| Middle HDI tertile | 0.60 (0.36, 1.00) (0.048) | 0.67 (0.42, 1.06) (0.087) | | | | |
| Lower HDI tertile | 0.51 (0.31, 0.84) (0.008) | 0.47 (0.30, 0.73) (0.001) | | | | |
| Higher GDP tertile (ref.) | | | | | | |
| Middle GDP tertile | | | 0.58 (0.41, 0.83) (0.003) | 0.68 (0.48, 0.95) (0.023) | | |
| Lower GDP tertile | | | 0.60 (0.45, 0.81) (0.001) | 0.54 (0.41, 0.71) (<0.001) | | |
| Higher Gini tertile (ref) | | | | | | |
| Middle Gini tertile | | | | | 1.28 (0.70, 2.33) (0.430) | 1.21 (0.75, 1.93) (0.439) |
| Lower Gini tertile | | | | | 1.07 (0.57, 2.03) (0.830) | 1.09 (0.66, 1.79) (0.747) |
| Loneliness (ref. never) | | | | | | |
| Rarely /sometimes | 0.92 (0.87, 0.97) (0.001) | 1.01 (0.95, 1.07) (0.766) | 0.94 (0.87, 1.02) (0.129) | 1.06 (0.96, 1.16) (0.258) | 0.93 (0.87, 1.00) 0.038 | 1.01 (0.93, 1.09) (0.889) |
| Most of the time/always | 1.08 (1.01, 1.16) (0.0280) | 1.03 (0.94, 1.13) (0.471) | 1.12 (1.01, 1.25) (0.034) | 1.07 (0.93, 1.24) (0.320) | 1.04 (0.95, 1.14) (0.430) | 0.95 (0.84, 1.07) (0.418) |
| Smoking days (ref. none) | | | | | | |
| 1–5 days | 1.08 (0.99, 1.17) (0.078) | 1.08 (0.99, 1.17) (0.075) | 1.14 (0.99, 1.31) (0.064) | 1.03 (0.90, 1.17) (0.696) | 1.01 (0.90, 1.13) (0.884) | 1.07 (0.95, 1.19) (0.259) |
| 6 or more days | 1.21 (1.09, 1.33) (<0.001) | 1.08 (0.98, 1.20) (0.117) | 1.31 (0.11, 1.55) (0.001) | 1.03 (0.88, 1.21) (0.710) | 1.07 (0.93, 1.24) (0.330) | 1.01 (0.89, 1.15) (0.855) |
| Alcohol drinking days (ref. none) | | | | | | |
| 1 or 2 days | 0.95 (0.89, 1.01) (0.078) | 0.87 (0.82, 0.93) (<0.001) | 1.00 (0.92, 1.10) (0.938) | 0.94 (0.85, 1.04) (0.22) | 0.93 (0.86, 1.00) (0.052) | 0.91 (0.84, 0.98) (0.019) |
| 3 or more days | 0.86 (0.80, 0.92) (<0.001) | 0.82 (0.77, 0.88) (<0.001) | 0.98 (0.87, 1.10) (0.702) | 0.94 (0.84, 1.04) (0.229) | 0.91 (0.83, 1.00) (0.050) | 0.87 (0.80, 0.95) (0.002) |
| Food insecurity (ref. never or sometimes) | | | | | | |
| Most of the time/always | 1.05 (0.99, 1.11) (0.111) | 1.10 (1.04, 1.17) (0.002) | 0.99 (0.91, 1.08) (0.817) | 0.95 (0.86, 1.04) (0.266) | 1.01 (0.94, 1.10) (0.710) | 1.06 (0.98, 1.15) 0.142 |
| *Intraclass Correlation Coefficient* | *3.4%* | *2.6%* | *4.1%* | *2.7%* | *3.9%* | *2.7%* |

CI, confidence interval; GDP, gross domestic product; HDI, human development index; OR, odds ratio.

obesity and suicidal ideation with planning, and their interconnectedness, that are amenable to policy and practice.

Overconsumption of high-energy foods and lack of physical activity are the primary behavioural risk factors for obesity, shaped by economic, social, and commercial determinants. The nutrition transition refers to a shift from diets composed of whole foods (e.g., pulses and whole grains) and low in refined oils and sugars to an energy-dense and nutrient-poor diet composed of fat and sugar-rich diets, and processed foods. The LAC countries experienced these dietary shifts earlier than most LMICs. The LAC countries are among the biggest consumers of sugar-sweetened beverages (SSBs) and fruit juices in the world, consuming more than triple the recommended WHO levels. In a recent global survey, average daily consumption of SSBs was highest among Mexicans (approximately 500 m/day), followed by Surinamese

**Table 5. Association between national indices of development and economic inequality and suicidal ideation with planning, adjusted for individual risk behaviours, by sex (N = 59,061).**

| | Girls | Boys | Girls | Boys | Girls | Boys |
|---|---|---|---|---|---|---|
| | OR (95% CI) (p-value) | OR (95% CI) (p-value) | OR (95% CI) (p-value) | OR (95% CI) (p-value) | OR (95% CI) (p-value) | OR (95% CI) (p-value) |
| Higher HDI tertile (ref.) | | | | | | |
| Middle HDI tertile | 0.80 (0.50, 1.27) (0.339) | 0.93 (0.50, 1.72) (0.684) | | | | |
| Lower HDI tertile | 0.89 (0.56, 1.41) (0.615) | 0.94 (0.73, 2.28) (0.376) | | | | |
| Highest GDP tertile (ref.) | | | | | | |
| Middle GDP tertile | | | 0.92 (0.60, 1.39) (0.681) | 0.79 (0.58, 1.10 (0.67) | | |
| Lower GDP tertile | | | 1.03 (0.76, 1.40) (0.829) | 0.96 (0.74, 1.25) (0.383) | | |
| Higher Gini tertile | | | | | | |
| Middle Gini tertile | | | | | 1.16 (0.68, 2.00) (0.681) | 1.00 (0.58, 1.72) (0.886) |
| Lower Gini tertile | | | | | 1.31(0.78, 2.22) (0.829) | 1,12 (0.66, 1.88) (0.676) |
| Loneliness (ref. never) | | | | | | |
| Rarely /sometimes | 1.82 (1.68, 1.97) (<0.001) | 1.71 (1.52, 1.92) (<0.001) | 1.82 (1.52, 1.92) (<0.001) | 1.71 (1.52, 1.92) (<0.001) | 1.90 (1.73, 2.08) (<0.001) | 1.79 (1.57, 2.05) (<0.001) |
| Most of the time/always | 5.18 (4.72, 5.69) (<0.001) | 4.97 (4.33, 5.69) (<0.001) | 5.17 (4.17, 5.68) (<0.001) | 4.87 (4.25, 5.80) (<0.001) | 5.78 (5.21, 6.42) (<0.001) | 5.09 (4.35, 5.94) (<0.001) |
| Smoking days (ref. none) | | | | | | |
| 1 to 5 days | 2.04 (1.82, 2.29) (<0.001) | 1.74 (1.50,2.03) (<0.001) | 2.06 (1.84, 2.31) (<0.001) | 1.99 (1.70, 2.34) (<0.001) | 2.11 (1.87, 2.38) (<0.001) | 2.00 (1.70, 2.34) (<0.001) |
| 6 or more days | 2.53 (2.21, 2.91) (<0.001) | 2.45 (2.07, 2.90) (<0.001) | 2.22 (2.03, 2.42) (<0.001) | 2.92 (2.45, 3.23) | 2.61 (2.62, 3.01) (<0.001) | 2.86 (2.39, 3.43) (<0.001) |
| Alcohol drinking days (ref. none) | | | | | | |
| 1 or 2 days | 1.88 (1.72, 2.06) (<0.001) | 1.51 (1.32, 1.73) (<0.001) | 1.85 (1.69, 2.03) (<0.001) | 1.46 (1.28, 1.67) (<0.001) | 1.86 (1.68, 2.06) (<0.001) | 1.52 (1.30, 1.77) (<0.001) |
| 3 or more days | 2.47 (2.23, 2.74) (<0.001) | 2.00 (1.73, 2.31) (<0.001) | 2.41 (2.17, 2.67) (<0.001) | 1.90 (1.65, 2.18) (<0.001) | 2.39 (2.13, 2.69) (<0.001) | 1.97 (1.69, 2.30) (<0.001) |
| Food insecurity (ref. never or sometimes) | | | | | | |
| Most of the time/always | 1.03 (0.94, 1.13) | 1.19 (1.06, 1.34) (<0.001) | 1.03 (0.94, 1.13) (0.540) | 1.06 (0.94, 1.21) (0.600) | 1.06 (0.96, 1.19) (0.580) | 1.09 (0.94, 1.27) (0.650) |
| *Intraclass correlation coefficient* | *2.2%* | *1.8%* | *2.5%* | *1.4%* | *3.0%* | *2.8%* |

CI, confidence interval; GDP, gross domestic product; HDI, human development index; OR, odds ratio.

and Jamaicans (approximately 400 ml). Information on dietary habits is patchy across the LAC context and often rely on sales data, but they signal that snacking has also become a major component of diets, an increase in eating away from home, and a reduction in consumption of legumes, fruits and vegetables, and whole grains [30,31].

Popkin and Reardon provide a cogent argument of the link between the transformation of food systems and dietary shifts globally [32] and also in the LAC [1]. They identified several factors pertinent to the LAC context, including income growth, policy liberalisation, and privatisation in the 1980s, improvement of infrastructure, urbanisation and women's work outside the home, the supermarket revolution, and reduced physical activity due to introductions of activity-saving technologies. These transformations begun in more economically

advantaged countries such as Brazil, Mexico, Barbados, Bahamas, and Trinidad and Tobago. The rapid transformations in some of the countries, for example, Guyana which shifted from low income to upper middle income over last 3 years, has been accompanied by simultaneous manifestation of both undernutrition and overweight and obesity.

The increased GDP and subsequent reduction of poverty in LMICs led to a surge in literature examining the interrelationships of growing national wealth, inequality, and health (mainly focused on mortality and NCD). Wilkinson and Pickett found that, regardless of national wealth, high income inequality at the country level was associated with higher levels of overweight [33]. Overweight has been traditionally considered to be more prevalent among the more advantaged in LMICs, but there is increasing evidence that the burden of obesity shifts to the poor as countries develop [34]. Monteiro and colleagues found that when GDP per capita per annum is <US$2,500, poverty is inversely associated with overweight; above this threshold, the burden of overweight shifts from the wealthy to the poor [35]. A positive association between socioeconomic circumstances and obesity in children, regardless of the level of gross national income (GNI) per capita, has been previously shown [36]. We found that food insecurity is associated with a lower likelihood of overweight for boys in poorer countries in the LAC region, signalling future increases in the prevalence of overweight among boys could be expected as countries transition with increases in national incomes.

Overweight/obesity was positively associated with suicidal ideation with planning, after adjusting for covariates, lending support to the findings of the review by Amiri and Behnezhad of a positive relationship between overweight/obesity and suicidal ideation with planning [4]. Overweight/obesity might be on the causal pathway from economic poverty to potentially suicidal ideation with planning (e.g., poverty leading to obesity and to suicide with ideation planning) or, alternatively, from economic poverty to psychosocial stress, suicidal thoughts, and unhealthy lifestyle behaviours. The cross-sectional design of the GSHS does not allow for testing these hypotheses.

National indices were not associated with suicidal ideation with planning. This partially aligns with work by Wilkinson and Pickett who found that average levels of country income were not associated with the 2013 UNICEF child well-being index, but that relative income inequality was associated with child well-being [24]. Contrary to the latter finding, our study did not find an association between relative income inequality (Gini index) and suicidal ideation with planning. The importance of psychosocial factors (e.g., loneliness and bullying) for both suicidal behaviours and overweight, and the stronger associations with suicidal behaviours, also aligns with other studies [37,38]. More than half of all young people in LAC countries, across varying levels of national development, can be considered at risk and confront a series of challenges including violence, poverty, and environmental hazards [1]. Multiple adversities in childhood increases the risk of suicidal behaviour [39,40]. Environmental hazards (climate change, natural catastrophe, conflict, and migration) are associated with depression, anxiety, and suicide [41]. The higher prevalence of suicidal ideation among girls than boys and higher levels of loneliness have been reported by several studies [25,42,43] Despite the higher prevalence of suicidal ideation among girls, there is a gender suicide paradox that should be acknowledged by future studies according to which higher proportions of boys commit suicide [44].

## Strengths and limitations

The study included 21 of 33 countries in the LAC with nationally representative samples. The GSHS uses standardised methods, data collection, and questionnaires that facilitate the

comparability of the data across countries. There are some limitations. Data were cross-sectional, which does not allow for causal interpretations.

Many children in LAC drop out of school and could be expected to have worse social and educational outcomes than those who stay in school [37]. These children are not captured GSHS. Information on important correlates of overweight/obesity, such as fizzy drink consumption and physical activity, was not included in the GSHS. The stigma and taboo around suicide might discourage young people to report suicidal ideation and planning, so the estimates should be considered as conservative. Data were collected in different years between 2009 and 2013, and a direct comparison of prevalence between countries should be made with caution. The bidirectional association between overweight/obesity and suicidal ideation with planning may influence our findings in addition to the confounding from unmeasured factors such as depression and mood disorders.

## Implications

Our findings provide evidence for policy and highlight specific research gaps. Given the health implications, there is an urgent need for surveillance systems to gather robust data that can be used to develop evidence-based policies that specifically target adolescent health. The omission of adolescent overweight and obesity from the SDGs is a major policy gap. The SDGs address poverty, education, nutrition, and universal health coverage and provide an opportunity for integrating policies that could coherently address the determinants of adolescent nutrition and mental health. Our findings support an ecological framework for policy interventions as well as further research on adolescent health to understand the complexities of multilevel factors at individual, household, community, and societal level to inform policy and practice interventions that are appropriate for the region. Given the significant variation in sociocultural and environmental contexts among countries, further research is needed to understand these behaviours. Our findings on the major global health problems of adolescent overweight/obesity and suicide-related behaviour, and their interconnectedness via structural and psychosocial factors, provide important evidence for policy makers.

## Supporting information

**S1 STROBE Checklist. STROBE checklist for observational studies.**
(DOC)

**S1 Analysis Plan.**
(DOCX)

**S1 Table. Distribution (%) of analysis's variables by country.**
(DOCX)

**S2 Table. Girls: Association between national indices of development, income inequality, and overweight/obesity, adjusted for individual risk factors.**
(DOCX)

**S3 Table. Boys: Association between national indices of development, income inequality, and overweight/obesity, adjusted for individual risk factors.**
(DOCX)

**S4 Table. Girls: Association between national indices of development, income inequality, and suicidal ideation with planning, adjusted for individual risk factors.**
(DOCX)

**S5 Table. Boys: Association between national indices of development, income inequality, and suicidal ideation with planning, adjusted for individual risk factors.**
(DOCX)

**S6 Table. Association between continuous national indices of development, income inequality, and z-BMI, adjusted for individual risk factors.** BMI, body mass index.
(DOCX)

**S7 Table. Association between continuous national indices, income inequality, and underweight and overweight/obesity relative to normal weight, adjusted for individual risk factors.**
(DOCX)

**S1 Fig. Age-adjusted predicted probabilities of overweight/obesity by HDI for boys and girls.** HDI, human development index.
(TIFF)

**S2 Fig. Age-adjusted predicted probabilities of overweight/obesity by Gini for boys and girls.**
(TIFF)

**S3 Fig. Age-adjusted predicted probabilities of suicide ideation with planning by GDP per capita ($) for boys and girls.** GDP, gross domestic product.
(TIFF)

**S4 Fig. Age-adjusted predicted probabilities of suicide ideation with planning by HDI for boys and girl.** HDI, human development index.
(TIFF)

**S5 Fig. Age-adjusted predicted probabilities of suicide ideation with planning by Gini for boys and girls.**
(TIFF)

## Author Contributions

**Conceptualization:** Seeromanie Harding.

**Data curation:** Seeromanie Harding.

**Formal analysis:** Christelle Elia, Seeromanie Harding.

**Funding acquisition:** Majella O'Keeffe, Ingrid Wolfe, Jane Sandall, Craig Morgan, J. Kennedy Cruickshank, Reeta Gobin, Rainford Wilks, Seeromanie Harding.

**Methodology:** Alexis Karamanos, Alexandru Dregan, Seeromanie Harding.

**Project administration:** Seeromanie Harding.

**Supervision:** Alexis Karamanos, Alexandru Dregan, Majella O'Keeffe, Ingrid Wolfe, Jane Sandall, Craig Morgan, J. Kennedy Cruickshank, Reeta Gobin, Rainford Wilks, Seeromanie Harding.

**Writing – original draft:** Christelle Elia.

**Writing – review & editing:** Alexis Karamanos, Alexandru Dregan, Majella O'Keeffe, Ingrid Wolfe, Jane Sandall, Craig Morgan, J. Kennedy Cruickshank, Reeta Gobin, Rainford Wilks, Seeromanie Harding.

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
