## [Editor Report · Decision Letter 0]

9 Mar 2020

Dear Dr Harding, 

Thank you for submitting your manuscript entitled "Macro level determinants and adolescent overweight and suicidal behaviours: Evidence from 20 Latin American and Caribbean Countries" for consideration by PLOS Medicine.

Your manuscript has now been evaluated by the PLOS Medicine editorial staff [as well as by an academic editor with relevant expertise] and I am writing to let you know that we would like to send your submission out for external peer review.

Kind regards,

Adya Misra, PhD,

Senior Editor

PLOS Medicine

---

## [Decision Letter · Decision Letter 1]

21 Jul 2020

Dear Dr. Harding,

Thank you very much for submitting your manuscript "Macro level determinants and adolescent overweight and suicidal behaviours: Evidence from 20 Latin American and Caribbean Countries" (PMEDICINE-D-20-00475R1) for consideration at PLOS Medicine. 

[LINK]

In light of these reviews, I am afraid that we will not be able to accept the manuscript for publication in the journal in its current form, but we would like to consider a revised version that addresses the reviewers' and editors' comments. Obviously we cannot make any decision about publication until we have seen the revised manuscript and your response, and we plan to seek re-review by one or more of the reviewers. 

We expect to receive your revised manuscript by Aug 11 2020 11:59PM. Please email us (plosmedicine@plos.org) if you have any questions or concerns.

We look forward to receiving your revised manuscript. 

Sincerely,

Adya Misra, PhD

Senior Editor 

PLOS Medicine

plosmedicine.org

Abstract

Please provide p values as needed

Please define LAC on first view

Limitations of your work should be included in a sentence at the end of the methods and findings section

Author summary

Throughout-please use square brackets for references and Vancouver style for bibliography

Introduction

On page 5 please revise “prevalence of undernutrition is highest in Guyana and income Trinidad with levels as high 19-20%” 

Did your study have a prospective protocol or analysis plan? Please state this (either way) early in the Methods section.

Methods

Please introduce GSHS, LMICs on first view

Results

Please include p values throughout, providing exact p values unless p<0.001

Please include the Discussion sub heading on Page 15

Please temper all assertions of primacy by adding “to our knowledge” or similar 

Page 15 please introduce SSB on first view 

Page 16 could you use an alternative to “better off countries” 

Please introduce NCDs on first view

Please ensure that the study is reported according to the STROBE guideline, and include the completed checklist as Supporting Information. When completing the checklist, please use section and paragraph numbers, rather than page numbers. Please add the following statement, or similar, to the Methods: "This study is reported as per the Strengthening the Reporting of Observational Studies in Epidemiology (STROBE) guideline (S1 Checklist)."

Please report your study according to the relevant guideline, which can be found here: http://www.equator-network.org/

The Data Availability Statement (DAS) requires revision. For each data source used in your study: 

Comments from the reviewers:

Reviewer #1: I confine my remarks to statistical aspects of this paper. The general approach is fine but I have some issues to resolve before I can recommend publication.

In the introduction, the authors use causal language where it is not appropriate, e.g. on the bottom of page 5. The relation between teen behaviors and parental relationships are complex and probably involve cause in both directions as well as from exogenous variables.

On p. 5 "fourth leading cause" is basically meaningless. It depends on how you label the causes. Give a rate (suicides per thousand per year or whatever),

p 6 I don't understand why Argentina was excluded. Having a large population is not a reason for exclusion. If you want to analyze data country by country, that's fine, but meta-analytic type methods are designed to deal with different sized samples.

p. 7 Do not divide GDP and HDI into tertiles. This throws away information and increases both type I and type II error, while decreasing the precision of estimates. Leave them continuous and use splines to investigate nonlinearity.

Table 2 BMI is a terrible measure of overweight and dichotomizing it makes it worse. See e.g. my blog post https://medium.com/peter-flom-the-blog/why-bmi-is-a-bad-measure-of-obesity-and-what-is-better-f8a62fc9ca49?source=friends_link&sk=4b2ea559ab12853beb577764f83d151a or many other citations. Maybe you can't get a good measure of obesity, but you could leave BMI continuous. Similarly, use age in years, times attacked and so on. Think about how you want to model the count variables, but treating them using optimal scaling seems reasonable. 

page 9 Please describe the weighting variable and how it was calculated. 

Peter Flom

Reviewer #2: Many thanks for inviting me to review this valuable contribution to the literature on the economic determinants of health. This cross-sectional study of an international survey finds increased GDP is associated with a reduced chance of adolescent overweight, but not suicide. The study provides a useful contribution in the Latin American and Caribbean setting. However, I have a few suggestions the authors may wish to consider that might improve their manuscript. 

Major issues

1. Exclusion of Argentina: It seems somewhat problematic to exclude Argentina from all analyses. I appreciate the authors concern that its inclusion could skew the results but I wonder if at least a sensitivity analysis could be included, given the likely policy interest within Argentina and the relative lack of analyses within Latin America more generally. 

2. Choice of macroeconomic exposures: An additional variable that might be worth investigating would be income inequality (e.g. measured through the Gini index - available through the UNU-WIDER database, for example)

3. Tertiles for the exposure: While I appreciate that tertiles are easy to interpret, this categorisation does seem slightly arbitrary and might reduce the statistical power for detecting associations. Would a sensitivity analysis treating the exposure as a continuous variable be worth considering?

4. Comparability of response rates: Some further information on the comparability of response rates across countries would be helpful. 

Minor issues

1. Measurement of body mass index: It wasn't clear to me how BMI was measured and whether there was any validation of the measures?

2. "A p-value of less than 0.05 was statistically significant" - I would suggest avoiding this binary approach to assessing associations. See: Sterne JAC, Cox DR, Smith GD. Sifting the evidence—what's wrong with significance tests?Another comment on the role of statistical methods. BMJ (Clinical research ed). 2001;322(7280):226-31.

3. Typos: Discussion "Individuals level factors such [as] age". "shifted from a low income to upper middle income [country]". "The higher prevalence [of] suicidal behaviour". 

Reviewer #3: This is an interesting paper, but I have some questions about the methods and suggestions about the presentation. I think one of the challenges is that there is so much included - there are two health outcomes, two measures of economic development that are potential explanatory factors, an interaction with food insecurity (which is listed as a covariate), the relationship between the two outcomes, all presented for males and females. This results in so many tables and graphs that much of the results are in supplementary tables. In light of this, I think the paper needs a rewrite, with a focus on the key research questions throughout, rather than trying to present everything.

1. The final sentence of the first paragraph is not referenced. 

2. Other than the fact that they are important outcomes for adolescents and young people, it is not clear enough from the beginning why you have focused on overweight and suicide. Although later in the Introduction you cite studies that show a relationship between them, you acknowledge later that this could be due to confounding. Is there a logical reason for studying these two outcomes together, for example is one on the causal pathway or the other?

3. The third paragraph of the Introduction is about malnutrition, undernutrition and excess weight. There is not a corresponding paragraph about suicide behaviour.

4. The fourth paragraph of the Introduction references papers that assessed the relationship between overweight or BMI and suicide behaviours. Were they looking for a linear association, or did they consider that there may be an increase in risk for both underweight and overweight?

5. In the sentence after refs 16&17, there is an 'and' missing between alcohol and tobacco.

6. In the Methods section, there is an 'and' missing on line 6.

7. In Table 2, BMI is categorised as 'normal weight' or 'overweight/obese'. This surprised me as, based on the Introduction, I thought you would be investigating underweight as well as overweight (and undernutrition is picked up in the Discussion again, so this seems to be quite an omission). Furthermore, the category 'overweight/obese' is referred to in the rest of the paper, including tables, as 'overweight', which is misleading.

8. Given the hierarchical nature of the data (individuals nested within countries) are the models fitted multilevel models? Please give more details, explaining how the methods used are appropriate to this data structure.

9. In the Statistical Analysis section, you introduce an interaction between the measures of development and food insecurity. This variable has not been mentioned in the Introduction, and a rationale is needed for testing this interaction and not others. The results do not appear in any of the tables - this may be one analysis too many.

10. I question the term 'suicidal behaviour' used throughout the paper. My understanding is that suicidal behaviour includes suicide and attempted suicide, and that the term 'suicidal ideation' - defined as thoughts about taking action to end one's life, including: identifying a method, having a plan, and/or having intent to act - is closer to the outcome you are using.

11. Figures 1 and 2 present 'predictive probabilities'. What are these predictions based on, and are they different to the observed rates? Please explain in the Methods section.

12. It is not clear, apart from lack of space, why results for suicidal behaviours are in supplementary tables while those for overweight are in the main paper, this feels a bit like selective reporting.

13. The fifth line from the bottom of p11, I'm afraid I can't see where 52%-62% comes from in the tables, and 45%-51% in the next line.

14. The results presented in the first two paragraphs of p12 are not entirely clear. For example, lines 2/3, frequent bullying and cigarette smoking was associated WITH a higher likelihood of overweight FOR GIRLS. The first two sentences of the second paragraph needs to be carefully checked. I think it says 'overweight' instead of 'suicidal behaviours' and the results do not seem to correspond entirely to those in the table.

15. The interaction presented at the end of the first paragraph on p12 is between food insecurity and lower v higher income countries. Is this according to the classifications in Table 1, or according to GDP/HDI as it says in the Statistical Analysis section? These results do not appear in any tables. 

16. If the section 'Overweight and suicide' is to be included, it follow logically from the aims and methods, so please make sure you explain why this is a useful analysis, and describe it in the 'Statistical Analysis' section. The results do not appear to be presented in a table. 

17. The first half of p15 are 'Main Findings', but after that it reads like the Discussion. 

18. P17, last sentence before 'Strengths and Limitations', should read 'the higher prevalence OF suicidal behaviour..' There is a well-recognised paradox here, that suicide is higher in males than females, which is not acknowledged in your Discussion, but deserves mention.

19. In the Stengths and Limitations, you mention that income inequality could not be found for all countries, might be worth adding to this sentence 'so was not included in the analysis'.

20. Figure 1. The country labels need to be included in a key. There is a missing footnote. Why is the y axis on the right rather than the left of the graphs.

21. Table 4. It is not clear why there are 4 columns rather than 2 (Boys and Girls). In places the columns appear to be repeated but this is not the case all the way through. Please give all Intraclass Correlation Coefficients to 1dp. It might be helpful to the reader to say that this is the 'Model 2' referred to in the paper.

22. Table 5. The row 'rarely/sometimes' has an entry for '192' which should be '1.92'. The row '1 to 5 days' is missing a comma in the first column of numbers. 

23. Supplementary Table 1. The title needs to be rephrased. It looks like you have presented percentages, but this is not stated. Please present all numbers to 1dp. It is not clear what the two numbers presented for M/F are, as they do not sum to 100%. 

24. Supplementary Table 2. . It might be helpful to the reader to say that this is the 'Model 1' referred to in the paper. Please change the group labels to read 'Higher GDP tertile', 'Middle GDP tertile' and 'Lower GDP tertile'. Similar for HDI, and this applies to Tables 4 & 5.

[LINK]

---

## [Decision Letter · Decision Letter 2]

7 Oct 2020

Dear Dr. Harding,

Thank you very much for re-submitting your manuscript "Macro level determinants and adolescent overweight and suicidal ideation with planning: Evidence from 21 Latin American and Caribbean Countries" (PMEDICINE-D-20-00475R2) for review by PLOS Medicine.

I have discussed the paper with my colleagues and the academic editor and it was also seen again by reviewers. I am pleased to say that provided the remaining editorial and production issues are dealt with we are planning to accept the paper for publication in the journal.

[LINK]

We look forward to receiving the revised manuscript by Oct 14 2020 11:59PM. 

Sincerely,

Adya Misra, PhD

Senior Editor 

PLOS Medicine

plosmedicine.org

Requests from Editors:

Please revise your title according to PLOS Medicine's style. Your title must be nondeclarative and not a question. It should begin with main concept if possible. "Effect of" should be used only if causality can be inferred, i.e., for an RCT. Please place the study design ("A randomized controlled trial," "A retrospective study," "A modelling study," etc.) in the subtitle (ie, after a colon).

Abstract-please provide brief participant demographics

Please remove "166 million young people" as most populous countries were not included and this can be misleading

Please remove the sentences about violence in the abstract and provide the background relevant to your current work

Please add to the abstract "sample sizes ranged from X for Anguilla to Y for Argentina", for example 

Please include the dominance of Argentinians in the dataset as a limitation

Abstract conclusions:* Please address the study implications without overreaching what can be concluded from the data; the phrase "In this study, we observed ..." may be useful. * Please interpret the study based on the results presented in the abstract, emphasizing what is new without overstating your conclusions. * Please avoid vague statements such as "these results have major implications for policy/clinical care". Mention only specific implications substantiated by the results. * Please avoid assertions of primacy ("We report for the first time....")

Please replace all iterations of “obese” with the phrase “with obesity” or “with overweight” throughout, in line with people first language principles

Please replace gender with “sex”

The author summary needs to be in bullet points please, with clear subsections. Please see our author guidelines for more information: https://journals.plos.org/plosmedicine/s/revising-your-manuscript#loc-author-summary

Please include all funding information in the article metadata only, this is not needed in the main text

Please use square brackets for references

I suggest rewording the introduction lines 71-78 to avoid repeating the text provided in the abstract

Please add details of GSHS data in the data availability statement, including a link to the dataset

Discussion- please discuss limitations of your study design, especially the bidirectional relationship between suicide ideation and overweight/obesity that may influence your findings in addition to the usual confounding factors. 

Please use paragraphs and sections in the STROBE checklist, as page numbers are likely to change 

Did your study have a prospective protocol or analysis plan? Please state this (either way) early in the Methods section.

Please add full access details for reference 19

Comments from Reviewers:

Reviewer #1: The authors have addressed my concerns and I now recommend publication

Reviewer #2: The authors appear to have done a good job in their revisions. One small outstanding suggestion I have is that it wasn't clear if BMI was measured or self-reported. I am assuming measured but I'm not sure it's explicitly stated - apologies if I've missed that.

[LINK]

---

## [Editor Report · Decision Letter 3]

9 Nov 2020

Dear Prof Harding, 

On behalf of my colleagues and the academic editor, Dr. Sanjay Basu, I am delighted to inform you that your manuscript entitled "Association of macro level determinants with adolescent overweight and suicidal ideation with planning: a cross-sectional study of 21 Latin American and Caribbean Countries" (PMEDICINE-D-20-00475R3) has been accepted for publication in PLOS Medicine. 

PRODUCTION PROCESS

Before publication you will see the copyedited word document (within 5 business days) and a PDF proof shortly after that. The copyeditor will be in touch shortly before sending you the copyedited Word document. We will make some revisions at copyediting stage to conform to our general style, and for clarification. When you receive this version you should check and revise it very carefully, including figures, tables, references, and supporting information, because corrections at the next stage (proofs) will be strictly limited to (1) errors in author names or affiliations, (2) errors of scientific fact that would cause misunderstandings to readers, and (3) printer's (introduced) errors. Please return the copyedited file within 2 business days in order to ensure timely delivery of the PDF proof. 

If you are likely to be away when either this document or the proof is sent, please ensure we have contact information of a second person, as we will need you to respond quickly at each point. Given the disruptions resulting from the ongoing COVID-19 pandemic, there may be delays in the production process. We apologise in advance for any inconvenience caused and will do our best to minimize impact as far as possible.

EARLY VERSION

PRESS

PROFILE INFORMATION

Thank you again for submitting the manuscript to PLOS Medicine. We look forward to publishing it. 

Best wishes, 

Adya Misra, PhD

Senior Editor 

PLOS Medicine

plosmedicine.org